# Enhancing Medical Image Generation with Anatomical Precision: A Multi-Headed VAE-Based Diffusion Model

## Abstract

Score-based image generation models, also known as diffusion models, can generate highly realistic and diverse natural images. However, a common challenge emerges when applying diffusion models to medical image generation and segmentation. While these models excel at producing realistic local textures, they struggle to accurately capture global anatomical priors, such as organ shape and location. Furthermore, the model lacks the capability for controlled recalibration to transform an anatomically unrealistic image into a realistic one. Here we present a new diffusion model where the generated images exhibit both realistic style and anatomically accurate position. Specifically, this is done by guiding the reverse diffusion process with our specially designed multi-headed VAE, which produces the image's disentangled style and position embeddings. We use the position embedding to define a grid deformation function that deforms a simple position prior to a predicted segmentation mask. Then, we apply the same grid deformation on the style embedding for image generation. This alleviates the style embedding from the burden of learning position features, thereby promoting disentangling. Our proposed approach showcases promising performance in controlled image generation across a range of medical image tasks, such as skin lesions and fetal head. Furthermore, our model delivers state-of-the-art segmentation performance.

## 1 Introduction

Variational autoencoders (VAEs) are a class of neural networks that model image representation and generation in a Bayesian approach (Kingma & Welling, 2013). Given an image, the encoder part of a VAE maps it to a low dimensional latent representation $z$, and the decoder part of a VAE try to reconstruct the image from $z$. One notable feature of VAE is the disentanglement of the latent variable $z$ (Higgins et al., 2017; Burgess et al., 2018; Dupont, 2018; Chen et al., 2018b). For image generation tasks, usually one semantic feature of images corresponds to a specific dimension of $z$. Varying this dimension of $z$ can generate new images that change only the corresponding semantic feature. This disentanglement provides users with some guidance over the image generation process. However, the correspondence between specific dimensions in $z$ and semantic features in images can only be discovered by varying individual dimensions of $z$ and manually checking the generated images. Another drawback of VAEs is that reconstructed images are usually blurry.

Denoising diffusion probabilistic models (DDPMs) are a class of latent variable generative models (Ho et al., 2020; Sohl-Dickstein et al., 2015) consisting of forward noising and backward denoising steps. Later Song et al. (2021) proposed a framework that unifies DDPMs as solving stochastic differential equations (SDEs) using the score-matching method. DDPMs can produce high-resolution and realistic images, but users have difficulty in guiding the generating process. This stems from the inherent nature of the diffusion based approach, which does not explicitly model low-dimensional semantic features.

In this work, we propose to build more meaningful image representation. VAEs can produce disentangled latent space, but randomness in the correspondence between feature dimensions and high level semantic features hinders usage of this disentanglement. We rectify the randomness problem

by decomposing latent space explicitly into position parameters and style parameters, so that users can easily analyze these aspects independently. This is done by designing a multi-headed VAE trained for both image generation and segmentation, where the segmentation task helps learn and isolate position information from image style information. We demonstrate the effectiveness of this decomposition by performing controlled image generation, where a user can update or interpolate either the style or the position parameters. Furthermore, this explicit disentanglement promotes more faithful and consistent refinement of blurry VAE outputs by using diffusion models.

The main contributions of our work can be summarized as follows:

1. **A novel VAE model for image segmentation and generation**: We propose a multi-headed VAE model for medical image segmentation and reconstruction. Instead of making pixel-wise predictions, our segmentation head uses a warping function to transform a reference mask into a segmentation mask. This warping function is also used to warp part of feature maps in the decoder to alleviate style parameters from learning position representation. We give details on the statistical reasoning of our model, and we show that our model achieves segmentation results comparable to other state-of-the-art methods.

2. **Meaningful latent representation with decoupled style and position**: Different from standard VAEs, our VAE decomposes the latent parameters into style and position parameters, where the position part controls the location and shape of the segmentation target. We demonstrate that manipulating style or position latent parameters will only affect the corresponding part of the generated images.

3. **Controllable realistic image generation**: We propose a conditional DDPM (cDDPM) combines our multi-headed VAE and a continuous conditional diffusion model, which can effectively refine blurry VAE reconstructions. Compared with DiffuseVAE (Pandey et al., 2022) which also combines a VAE with a DDPM model, our cDDPM can give more faithful reconstructions regarding both image style and segmentation region. Moreover, users can easily update style and position parameters and guide the reconstruction process.

## 2 BACKGROUND

### 2.1 VARIATIONAL AUTOENCODERS (VAEs)

VAEs are encoder-sampling-decoder structured neural networks that model image generation in a Bayesian approach. Given a latent variable $z \overset{p}{\sim} \mathcal{N}(\mathbf{0}, \mathbf{1})$, one assumes an observed data sample is generated from $x|z \overset{p_\phi}{\sim} \mathcal{N}(x; \mu, \sigma^2)$, where $(\mu, \sigma)$ is calculated by a decoder network $\mathrm{DEC}_\phi(z)$ with learnable parameters $\phi$. On the other hand, given a data sample $x$, the posterior probability of observing the latent $z$ is *approximated* by $q_\psi(z|x) = \mathcal{N}(z; \tilde{\mu}, \tilde{\sigma}^2)$ where $(\tilde{\mu}, \tilde{\sigma})$ is calculated by an encoder network $\mathrm{ENN}_\psi(x)$ with learnable parameters $\psi$. The sampling part of a VAE performs the sampling of $z|x$ given the encoder outputs.

The encoder and decoder are trained jointly by maximizing the evidence lower bound (ELBO) function (Blei et al., 2017) defined by

$$\mathcal{L}(\phi, \psi|x) = \mathbb{E}_{q_\psi(z|x)}[\log p_\phi(x|z)] - D_{\mathrm{KL}}(q_\psi(z|x)\|p(z)), \tag{1}$$

which bounds from below the marginal log-likelihood $\log p_\phi(x)$. By utilizing a reparametrization technique introduced by Kingma & Welling (2013), one can easily minimize

$$\mathbb{E}_{x \sim p_{\mathrm{data}}}[-\mathcal{L}(\phi, \psi|x)] \tag{2}$$

by using stochastic gradient descent methods.

### 2.2 DENOISING DIFFUSION PROBABILISTIC MODELS (DDPMs)

DDPMs are a class of latent variable generative models that consist of forward noising steps and backward denoising steps. Given a data sample $x_0$, the forward process is modeled by a first-order Markov chain with transition probability

$$q(x_t|x_{t-1}) = \mathcal{N}(\sqrt{1 - \tilde{\beta}_t} x_{t-1}, \tilde{\beta}_t I), \quad t = 1, 2, \cdots, N, \tag{3}$$

where $\tilde{\beta} > 0$ is a monotonically increasing sequence. It can be shown that

$$q(\boldsymbol{x}_t|\boldsymbol{x}_0) = \mathcal{N}(\sqrt{\bar{\alpha}_t}\boldsymbol{x}_0, (1 - \bar{\alpha}_t)\boldsymbol{I}), \quad \bar{\alpha}_t = \prod_s (1 - \tilde{\beta}_s). \tag{4}$$

Intuitively, $\boldsymbol{x}_t|\boldsymbol{x}_0$ gradually loses information of $\boldsymbol{x}_0$ and mimics isotropic Gaussian noise as $t$ increases. When $N$ is large and $\tilde{\beta}_t$ is well-behaved, the forward Markov process can be reversed, where the backward transition probability can be approximated by

$$p(\boldsymbol{x}_{t-1}|\boldsymbol{x}_t) = \mathcal{N}(\boldsymbol{x}_{t-1}; \boldsymbol{\mu}_t, \sigma_t^2 \boldsymbol{I}), \quad \boldsymbol{x}_N \sim \mathcal{N}(\boldsymbol{0}, \boldsymbol{I}). \tag{5}$$

Here $\sigma_t$ can be either approximated by $\tilde{\beta}_t$ or by $\dfrac{1 - \bar{\alpha}_{t-1}}{1 - \bar{\alpha}_t}\tilde{\beta}_t$. The unknown mean vector $\boldsymbol{\mu}_t$ can be approximated by a temporal neural network $\tilde{\boldsymbol{\mu}}_{\boldsymbol{\xi}}(\boldsymbol{x}_t, t)$. When $\tilde{\boldsymbol{\mu}}_{\boldsymbol{\xi}}$ is properly trained, a data point can be sampled by first sample an isotropic Gaussian noise $\boldsymbol{x}_N$, and then follow the Markov transitions in equation 5.

It is shown in Song et al. (2021) that the forward noising steps in equation 3 is a discretization of the stochastic differential equation (SDE)

$$\mathrm{d}\boldsymbol{x} = -\frac{1}{2}\beta(t)\boldsymbol{x}\mathrm{d}t + \sqrt{\beta(t)}\mathrm{d}\boldsymbol{w}, \qquad \boldsymbol{x}(0) \sim p_{\text{data}}, \quad t \in [0, 1], \tag{6}$$

where the variance scheduler $\tilde{\beta}_i$ in equation 3 is $\dfrac{\beta(i/N)}{N}$, and $\boldsymbol{w}$ is the standard Brownian motion. It can be shown that the random variable $\boldsymbol{x}(t)$ converges to isotropic Gaussian noise as $t \to 1$. The SDE in equation 6 can be reverted by the reverse SDE

$$\mathrm{d}\boldsymbol{x} = \left[-\frac{1}{2}\beta(t)\boldsymbol{x} - \beta(t)\nabla_x \log p_t(\boldsymbol{x})\right]\mathrm{d}t + \sqrt{\beta(t)}\mathrm{d}\bar{\boldsymbol{w}}, \tag{7}$$

where $t$ goes backward from 1 to 0, $\mathrm{d}t$ is a negative infinitesimal time-step, $\bar{\boldsymbol{w}}$ is the standard Brownian motion running backward in time, and $p_t$ is the probability density function of $\boldsymbol{x}(t)$ in the forward equation 6, which involves data distribution and is unknown. By starting from the random variable $\boldsymbol{x}(1)$ as given by equation 6, which can be well approximated by an isotropic Gaussian noise, and following equation 7 we can recover the random variable $\boldsymbol{x}(0)$ that follows the data distribution. The discrete backward denoising step in equation 5 is a special discretization strategy for solving the backward SDE 7.

The function $\nabla_x \log p_t(\boldsymbol{x})$, which is the only missing part in solving equation 7, is called the score function of the random variable $\boldsymbol{x}$. It is proposed by Song et al. (2021) to use a temporal neural network $S_{\boldsymbol{\xi}}$ to approximate the score function by minimizing the loss function

$$\mathbb{E}_t \left\{\lambda(t)\mathbb{E}_{\boldsymbol{x}(0)\sim p_{\text{data}}}\mathbb{E}_{\boldsymbol{x}(t)|\boldsymbol{x}(0)} \left[||S_{\boldsymbol{\xi}}(\boldsymbol{x}(t), t) - \nabla_x \log p(\boldsymbol{x}(t)|\boldsymbol{x}(0))||_2^2\right]\right\}, \tag{8}$$

where $\lambda(t)$'s are weighting scalars inversely proportional to $\mathbb{E}[||\nabla_x \log p(\boldsymbol{x}(t)|\boldsymbol{x}(0))||_2^2]$. Once $S_{\boldsymbol{\xi}}$ is properly trained, one can substitute the score function in equation 7 by $S_{\boldsymbol{\xi}}$ and employ a numerical solver to solve the SDE.

## 2.3 WHEN DIFFUSION MODELS MEET VARIANTIONAL AUTO-ENCODERS

Data generation using DDPMs (either continuous or discrete) is actually data sampling, and it is difficult to have fine-tuned control over the sampling process. On the other hand, one has some control over VAE generating process because VAEs have disentangled latent spaces that corresponds to humanly understandable semantic features. To fuse these two methods, Pandey et al. (2022) proposed DiffuseVAE models which modified the forward transition steps in equation 3 to

$$q(\boldsymbol{x}_t|\boldsymbol{x}_{t-1}, \hat{\boldsymbol{x}}_0) = \mathcal{N}(\sqrt{1 - \tilde{\beta}_t}\boldsymbol{x}_{t-1} + \gamma_t\hat{\boldsymbol{x}}_0, \tilde{\beta}_t\boldsymbol{I}), \quad \gamma_t = 1 - \mathbb{1}_{t=1}\sqrt{1 - \tilde{\beta}_t}, \tag{9}$$

where $\boldsymbol{x}_0$ is a data sample, $\mathbb{1}_{t=1}$ is the indicator function for $t = 1$, and $\hat{\boldsymbol{x}}_0$ is a reconstruction from a trained VAE for $\boldsymbol{x}_0$. For large $t$, $\boldsymbol{x}_t|\hat{\boldsymbol{x}}_0$ approximately follows $\hat{\boldsymbol{x}}_0 + \boldsymbol{\varepsilon}$ for isotropic Guassian noise $\boldsymbol{\varepsilon}$.[1] Instead of starting from isotropic Gaussian noise, now the backward transitions start from $\hat{\boldsymbol{x}}_0$ with added Gaussian noise. By providing specific $\hat{\boldsymbol{x}}_0$ users give guidance on the sampling process. We refer readers to the original paper for backward transition probabilities.

---

[1]This is formulation 2 in (Pandey et al., 2022).

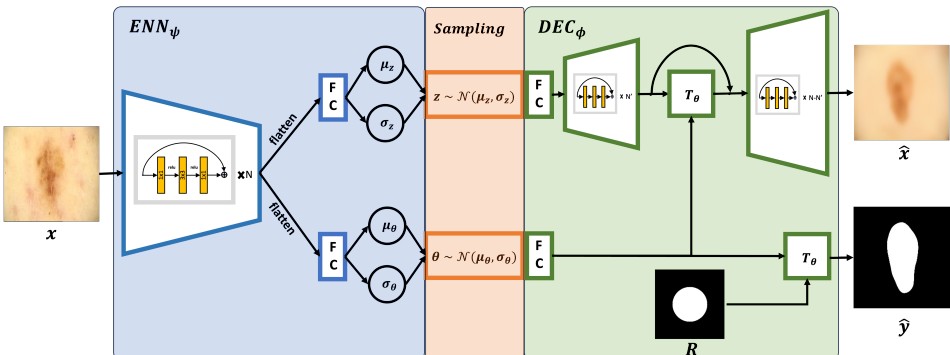

Figure 1: Illustration of Multi-headed VAE. $\text{ENN}_\psi$ takes an observed image $x$ and estimates the means and standard deviations of $z|x$ and $\theta|x$, assumed to follow normal distribution. Given sampled $z$ and $\theta$, $\text{DEC}_\phi$ generates reconstructed image $\hat{x}$ and predicted mask $\hat{y}$. We apply warping function $\text{T}_\theta$ on prior mask $R$ to generate the predicted mask. Likewise, when generating the reconstructed image, we apply the same warping function but only partially.

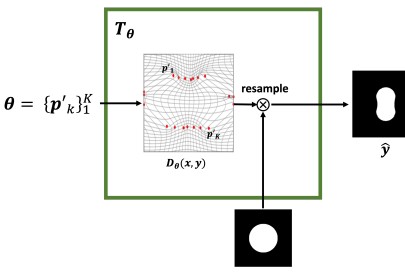

Figure 2: A TPS warping function $\text{T}_\theta$ takes position parameter $\theta$ and prior mask $R$ as inputs. Position parameter, denoted as red dots, is used to construct a deformed grid $\mathcal{D}_\theta(x, y)$. Thereafter, a differentiable resampling function is used to resample pixel intensities of the reference mask $R$. The same TPS deformation is applied to a part of upsampled style embedding to promote the disentanglement of style/position.

## 3  MULTI-HEADED VAE

In this work, we are mainly interested in analyzing medical data $(x, y)$, where $x$ is an image containing an object of interest (e.g. an organ or a lesion), and $y$ is a segmentation mask of the object. We assume the object has a prior shape that can be represented by a reference mask $R$. In this section, we describe the structure of our multi-headed VAE, which extends VAEs as in section 2.1 to both image generation and segmentation. In Appendix A we detail the statistical reasoning for the design of our multi-headed VAE and the loss function.

Our multi-headed VAE also has the encoder-sampling-decoder structure, where we use deep neural networks to model encoder $\text{ENN}_\psi$ and decoder $\text{DEC}_\phi$. For our purpose, we use two sets of parameters, the *style parameter* $z$ and the *position parameter* $\theta$, to jointly model the latent variable of our model. We assume both parameters have standard normal distributions as prior distributions and they are independent.

As shown in figure 1, $\text{ENN}_\psi$ takes an observed image $x \in \mathbb{R}^{h \times w \times c}$ as input to estimate the means and standard deviations of $z|x$ and $\theta|x$, where we assume the posteriors are also normal. During the sampling process, to ensure differentiability, we sample $z$ and $\theta$ using the reparameterization trick introduced in (Kingma & Welling, 2013). Given sampled $z$ and $\theta$, $\text{DEC}_\phi$ generates reconstructed image $\hat{x} \in \mathbb{R}^{h \times w \times c}$ and predicted mask $\hat{y} \in \{0, 1\}^{h \times w}$. To ensure $z$ only represents style of images and $\theta$ only represents position of segmentation objects, the decoder needs to be carefully designed.

### 3.1  ENCODER

Our $\text{ENN}_\psi$ consists of 3 sub-modules – ConvNet and two fully connected (FC) modules. ConvNet is a standard convolution neural network such as ResNet (He et al., 2015) that encodes an image into feature maps. FC module is a concatenation of multiple fully-connected and leakyReLU layers that projects the image embedding to the means and standard deviations of $\theta$ and $z$ after flattening.

## 3.2 DECODER – SEGMENTATION HEAD

As shown in figure 1, the segmentation head in our multi-headed VAE maps the estimated position posterior sample $\boldsymbol{\theta}|\boldsymbol{x}$ to a segmentation mask $\hat{\boldsymbol{y}}$. Instead of taking the usual ConvNet-based segmentation approach, which models the pixel-wise class attribute probabilities, we regard image segmentation as a spatial operation that deforms a disc $\boldsymbol{R}$ to a more complex shape. We achieve this by adopting spatial transformer network (STN) (Jaderberg et al., 2015) as our segmentation head, where the transformation is done by a warping function $T_{\boldsymbol{\theta}}$.

As shown in figure 2, warping function $T_{\boldsymbol{\theta}}$ takes position parameter $\boldsymbol{\theta}$ and prior mask $\boldsymbol{R}$ as inputs. Position parameter is used to construct a deformed grid $\mathcal{D}_{\boldsymbol{\theta}}(x, y)$, which serves as a function that maps target coordinate $\mathbf{p} = (x, y)$ (coordinate in predicted mask $\hat{\boldsymbol{y}}$) to a source coordinate $\mathbf{p}' = (x', y')$. Thereafter, a differentiable resampling function, such as bilinear interpolation, is used to resample pixel intensities of the reference mask $\boldsymbol{R}$ according to the deformed grid $\mathcal{D}_{\boldsymbol{\theta}}(x, y)$, e.g.

$$\hat{\boldsymbol{y}}(x, y) = T_{\boldsymbol{\theta}}(\boldsymbol{R})(x, y) = \text{resample}(\boldsymbol{R}, \mathcal{D}_{\boldsymbol{\theta}}(x, y)) = \text{resample}(\boldsymbol{R}, (x', y')). \tag{10}$$

Any differentiable coordinate deformation function can be considered. In this paper, we experiment with two deformation functions: 2D affine transform and thin plate spline (TPS) transform.

### 3.2.1 AFFINE TRANSFORM

Given a position parameter $\boldsymbol{\theta}$, the affine transform is defined as

$$\begin{bmatrix} \mathbf{p}' \\ 1 \end{bmatrix} = \begin{bmatrix} \theta_{11} & \theta_{12} & \theta_{13} \\ \theta_{21} & \theta_{22} & \theta_{23} \\ 0 & 0 & 1 \end{bmatrix} \begin{bmatrix} \mathbf{p} \\ 1 \end{bmatrix}. \tag{11}$$

Affine transform allows rotation, translation, scaling, and reflection. In actual implementation, the coordinates for the predicted mask are normalized such that $-1 \le x, y \le 1$. The reference mask $\boldsymbol{R}$ is padded with $0$ for source coordinates outside $[-1, 1]$.

While the affine transform is simple to implement and only requires estimating 6 position parameters, it only generates a linearly deformed grid. Consequently, when $\boldsymbol{R}$ is a disc, the resulting segmentation mask always has an elliptical shape. Therefore, we can only use affine transforms in our segmentation head when the actual masks exhibit very strong shape priors.

### 3.2.2 THIN PLATE SPLINE TRANSFORM

We propose to use TPS deformations (Bookstein, 1989) to allow non-linear deformations. Given a sequence of target control points $(\mathbf{p}_1, \cdots, \mathbf{p}_K)$ and their corresponding source control points $(\mathbf{p}'_0, \cdots, \mathbf{p}'_K)$, a unique TPS deformation function $\mathcal{D}$ is determined by matching the control points $\mathbf{p}_i \mapsto \mathbf{p}'_i$ with minimal bending energy. The matching of the $x$ and $y$ coordinates of the control points, together with 6 regularization conditions, give two sets of coefficients $(a_1, \cdots, a_{K+3})$ and $(b_1, \cdots, b_{K+3})$, such that a general target point $\mathbf{p} = (x, y)$ is mapped to $(\mathcal{D}_x(\mathbf{p}), \mathcal{D}_y(\mathbf{p}))$ with

$$\mathcal{D}_x(\mathbf{p}) = a_{K+1} + a_{K+2}x + a_{K+3}y + \sum_{k=1}^{K} a_k U(|\mathbf{p} - \mathbf{p}_k|), \tag{12a}$$

$$\mathcal{D}_y(\mathbf{p}) = b_{K+1} + b_{K+2}x + b_{K+3}y + \sum_{k=1}^{K} b_k U(|\mathbf{p} - \mathbf{p}_k|), \tag{12b}$$

where $U(r) = r^2 \log r^2$ is a kernel function.

For our implementation, note that the position parameter $\boldsymbol{\theta}$ consists of coordinates of the source control points, and we use a fixed set of target control points. Application of a TPS transform in a warping function $T_{\boldsymbol{\theta}}$ is shown in figure 2. Further details on TPS are given in Appendix B.

### 3.3 DECODER – IMAGE GENERATION HEAD

As illustrated in figure 1, our image generation head reconstructs the image $\hat{\boldsymbol{x}}$ from both the style posterior sample $\boldsymbol{z}|\boldsymbol{x}$ and the position posterior sample $\boldsymbol{\theta}|\boldsymbol{x}$.

The structure of the image generation head is similar to that of other convolutional VAEs, with the key distinction being the injection of the position posterior into an intermediate layer. This alleviates the style parameter from the burden of learning position features, thereby promoting disentanglement between the style and position parameters.

image generation head first maps the style parameter to a spatial size of $32 \times 32$ with a series of 2D convolutional and bilinear upsampling modules. Similar to the approach employed in the segmentation head, we also warp the upsampled style parameter using the learned position parameter $\boldsymbol{\theta}$. However, in contrast to the warping function in the segmentation head, which deforms the entire channel of the prior mask $\boldsymbol{R}$, we apply deformation exclusively to the first half of the upsampled style parameter. We assume that only half of the channel is sufficient for representing the object of interest, which we intend to deform. This partial warping of style parameter is shown in figure 1 as a skip connection around warping function $\mathrm{T}_{\boldsymbol{\theta}}$. Lastly, we map the partially deformed image feature to the reconstructed image $\hat{\boldsymbol{x}}$ using additional convolutional and upsampling modules.

### 3.4 LOSS FUNCTION

The loss function for the training of multi-headed VAE has three parts. For image reconstruction, we use the log-cosh loss function proposed by Chen et al. (2019) to measure the similarity between true image $\boldsymbol{x}$ and the reconstructed image $\hat{\boldsymbol{x}}$. For segmentation mask prediction, we use the commonly used Dice score to compare true mask $\boldsymbol{y}$ and predicted mask $\hat{\boldsymbol{y}}$. The third part of the loss function measures the KL divergence between standard unit normal distribution with the distribution of $(\boldsymbol{\theta}, \boldsymbol{z})|\boldsymbol{x}$, which is a normal distribution with mean and variance predicted by the encoder (see figure 1).

## 4 CONDITIONAL REFINEMENT OF VAE OUTPUTS

Reconstructed images from variational auto-encoders are usually blurry (Kingma & Welling, 2013; Pandey et al., 2022) (see figure 3). We follow the DiffuseVAE approach in (Pandey et al., 2022) to propose a conditional denoising diffusion probabilistic model (cDDPM) with a modified continuous diffusion process to refine our VAE outputs.

### 4.1 FORWARD DIFFUSION PROCESS

Let $\hat{\boldsymbol{x}}$ be the reconstruction by our VAE model of an image $\boldsymbol{x_0}$. For the forward diffusion process, we use the stochastic differential equation (SDE) given by

$$\mathrm{d}\boldsymbol{x} = -\frac{1}{2}\beta(t)(\boldsymbol{x}(t) - \hat{\boldsymbol{x}})\mathrm{d}t + \sqrt{\beta(t)}\mathrm{d}\boldsymbol{w}, \quad \boldsymbol{x}(0) = \boldsymbol{x_0}, \tag{13}$$

where $t$ ranges from 0 to 1, $\boldsymbol{w}$ is the standard Brownian motion, and $\beta(t)$ is a noise scale function that is positive and monotonically increasing [2].

Conditional on the initial image $\boldsymbol{x_0}$, at time $t$ the random variable $\boldsymbol{x}|\boldsymbol{x_0}$ follows the normal distribution $\mathcal{N}(\boldsymbol{\mu}(t), \boldsymbol{\sigma}(t))$ where [3]

$$\boldsymbol{\mu}(t) = \hat{\boldsymbol{x}} + (\boldsymbol{x_0} - \hat{\boldsymbol{x}}) \cdot \exp\left(-\frac{1}{2}\int_0^t \beta(s)\mathrm{d}s\right), \quad \boldsymbol{\sigma}^2(t) = \boldsymbol{I} - \boldsymbol{I} \cdot \exp\left(-\int_0^t \beta(s)\mathrm{d}s\right). \tag{14}$$

We note that $\boldsymbol{x}|\boldsymbol{x_0}$ mimics the initial image $\boldsymbol{x_0}$ for small $t$, and it mimics $\hat{\boldsymbol{x}} + \boldsymbol{\varepsilon}$ for large $t$ where $\boldsymbol{\varepsilon}$ is an isotropic Gaussian noise.

### 4.2 BACKWARD DIFFUSION PROCESS

The reverse SDE of the forward equation 13 is given by

$$\mathrm{d}\boldsymbol{x} = \left[-\frac{1}{2}\beta(t)(\boldsymbol{x}(t) - \hat{\boldsymbol{x}}) - \beta(t)\nabla_x \log p_t(\boldsymbol{x})\right]\mathrm{d}t + \sqrt{\beta(t)}\mathrm{d}\overline{\boldsymbol{w}}, \tag{15}$$

---

[2]This SDE can be regarded as a continuous model of the form-2 DiffuseVAE in (Pandey et al., 2022), but with a different initial condition.

[3]Technically speaking, the VAE reconstruction $\hat{\boldsymbol{x}}|\boldsymbol{x_0}$ follows a posterior distribution, and thus $\boldsymbol{x}$ should be conditioned on both $\boldsymbol{x_0}$ and $\hat{\boldsymbol{x}}|\boldsymbol{x_0}$. Here, we abuse the notation to keep symbols manageable.

| Models | Backbone | ISIC_2018 | | RIM_CUP | | RIM_DISC | | FETAL | |
|---|---|---|---|---|---|---|---|---|---|
| | | HD($\downarrow$)±sd | Dice($\uparrow$)±sd | HD($\downarrow$) | Dice($\uparrow$) | HD($\downarrow$) | Dice($\uparrow$) | HD($\downarrow$)±sd | Dice($\uparrow$)±sd |
| UNet++ (Zhou et al., 2018) | ResNet50 | $24.06 \pm 0.80$ | $0.89 \pm 0.01$ | 22.25 | 0.77 | 11.79 | 0.96 | $9.31 \pm 1.78$ | $0.97 \pm 0.00$ |
| DeepLabV3+ (Chen et al., 2018a) | ResNet50 | $20.80 \pm 0.76$ | $\mathbf{0.90} \pm 0.01$ | 22.25 | 0.77 | 11.27 | 0.96 | $6.04 \pm 0.57$ | $0.97 \pm 0.00$ |
| UNETR (Hatamizadeh et al., 2022) | ViT-B/16 | $20.05 \pm 0.76$ | $\mathbf{0.90} \pm 0.01$ | 18.98 | 0.78 | 10.95 | 0.96 | $6.19 \pm 0.34$ | $0.97 \pm 0.00$ |
| UnetrSwin (Hatamizadeh et al., 2021) | Swin-B | $18.16 \pm 0.77$ | $\mathbf{0.90} \pm 0.00$ | 17.42 | 0.79 | 10.55 | 0.96 | $5.17 \pm 0.58$ | $\mathbf{0.98} \pm 0.00$ |
| TransUnet (Chen et al., 2021) | ResNet50+ViT-B/16 | $16.34 \pm 0.73$ | $\mathbf{0.90} \pm 0.00$ | 19.41 | 0.79 | 11.81 | 0.96 | $4.52 \pm 0.19$ | $\mathbf{0.98} \pm 0.00$ |
| FCSN (Jeon et al., 2022) | DResNet50 | $20.21 \pm 0.88$ | $0.88 \pm 0.01$ | 18.15 | 0.77 | 9.85 | 0.96 | $6.58 \pm 0.37$ | $0.97 \pm 0.00$ |
| Multi-headed VAE (Affine) | ResNet50 | - | - | **16.57** | **0.79** | **9.52** | 0.96 | 4.49±0.36 | $0.97 \pm 0.00$ |
| Multi-headed VAE (TPS) | ResNet50 | 16.32±0.54 | $0.89 \pm 0.00$ | 17.95 | **0.79** | 9.99 | 0.96 | $4.66 \pm 0.34$ | $0.97 \pm 0.00$ |
| Multi-headed VAE (TPS) | ResNet50+ViT-B/16 | $\mathbf{15.47} \pm 0.65$ | $\mathbf{0.90} \pm 0.00$ | 17.40 | 0.78 | 10.16 | 0.96 | $\mathbf{4.40} \pm 0.25$ | $\mathbf{0.98} \pm 0.00$ |

Table 1: Comparison of Dice and Hausdorff distance (HD) metrics between Multi-headed VAE and other semantic segmentation models. The standard deviation (sd) is calculated from 5-fold results, with the best results indicated in **bold**. Second best HD results are in blue. A lower Hausdorff score is better, while a higher Dice score is better.

where $t$ runs backward from 1 to 0, $dt$ is a negative infinitesimal time-step, $p_t$ is the distribution of the random variable $x$ in the forward diffusion equation 13, and $\overline{w}$ is the standard Brownian motion running in backward time. The initial random variable $x(1)$ has the form $\hat{x} + \varepsilon$, where $\hat{x}$ follows our VAE output distribution and $\varepsilon$ is an isotropic Gaussian noise. Solving equation 15 gives samples from data distribution where the conditional probability of observing our starting sample $\hat{x}$ is high.

Sampling $\hat{x} + \varepsilon$ is straight-forward since sampling $\hat{x}$ only requires sampling from priors (both normal distributions) of $\theta$ and $z$, passing through the decoder of our VAE, and another sampling from normal distribution (see Appendix A for details). We can use equation 15 to perform either pure sampling (see Appendix G) or reconstruction from a given VAE output.

As in (Song et al., 2021), we use score-matching to estimate the score function $\nabla_x \log p_t(x)$ by using a UNet-shaped temporal neural network $S(x, t)$. After $S$ is fully optimized, we solve the backward equation 15 by using Euler-Maruyama discretization, where we substitute $S$ for the true score function.[4]

## 5 EXPERIMENTS

### 5.1 SEGMENTATION PERFORMANCE

As shown in Table 1, we compare the segmentation performance of our model with other segmentation models on three datasets: ISIC_2018 (Codella et al., 2019), RIM-ONE-DL (Batista et al., 2020), and FETAL (van den Heuvel et al., 2018). The comparing models are broadly categorized into three model kinds: pure ConvNets including UNet++(Zhou et al., 2018) and DeepLabV3+(Chen et al., 2018a), transformer-based models such as UNETR (Hatamizadeh et al., 2022), Unetr-Swin (Hatamizadeh et al., 2021), TransUnet (Chen et al., 2021), and the shape prediction model FCSN (Jeon et al., 2022) with the DResNet (Yu et al., 2017) as the backbone. We employ the Dice score and Hausdorff Distance (HD) to assess the segmentation accuracy in our experiments. Although the Dice score is a widely adopted segmentation metric, it is not ideal for evaluating shape correspondence. In contrast, HD is more sensitive to shape correspondence, as it quantifies the maximum dissimilarity distance between objects. We perform 5-fold cross-validation for ISIC_2018 and FETAL since their test data do not have publicly available segmentation masks.

We use a disc at the center of an image with a radius of a quarter of the height of the image as the reference mask **R**. The real masks for ISIC_2018 has a diverse range of shapes that cannot be approximated by ellipses, so affine transforms clearly fail here and we only tested the TPS deformations on this dataset. A discussion on possible choices of control points for TPS transforms is in Appendix C.

Our proposed model significantly outperforms all the compared models in terms of the HD metric. We notice that for tasks that assume strong object priors, like FETAL, RIM_CUP, and RIM_DISC, affine deformation positively constrains the mask prediction. Additionally, we observe that the

---

[4]See codes at https://anonymous.4open.science/r/codes–anonymous–B71F for detailed implementation.

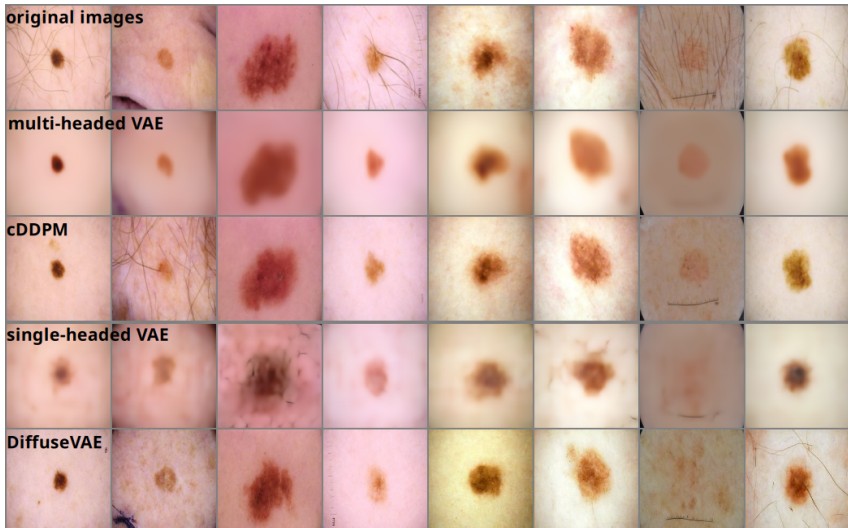

Figure 3: Reconstruction examples. First row: original validation images from ISIC. Second and third rows: our multi-headed VAE and cDDPM; fourth and fifth rows: single-headed VAE and DiffuseVAE (Pandey et al., 2022).

choice of backbone has an impact on performance, with a higher segmentation performance when using a transformer encoder compared to a pure ConvNet.

## 5.2 VISUALIZATIONS

In this subsection, we visualize images generated by our multi-headed VAE and cDDPM. We demonstrate that (i) our VAE can well preserve image styles and positions for objects of interest, and our cDDPM can provide faithful refinements of our VAE outputs, (ii) our VAE can successfully decompose style and position parameters, and (iii) users can easily provide guidance on the generating process.

### 5.2.1 RECONSTRUCTION RESULTS

In figure 3, we give reconstructions of our VAE (second row) and cDDPM (third row) on validation images from ISIC dataset. Our VAE outputs can give accurate descriptions of the lesion region and overall color tone of the original images, while our cDDPM can fill in details to the blurry VAE outputs. On the other hand, reconstructions from the single-headed VAE and the DiffuseVAE from (Pandey et al., 2022) cannot preserve lesion regions.

In figure 4 we give repeated reconstructions of 2 images (3rd and 8th images in figure 3) for both our cDDPM and the DiffuseVAE. Our method gives consistent reconstructions in terms of the lesion region and overall color tone, while local textures vary across samples (image can be enlarged for details). DiffuseVAE fails to preserve lesion regions across different reconstructions.

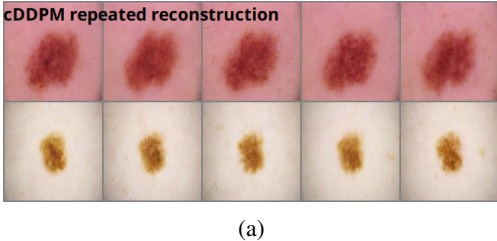

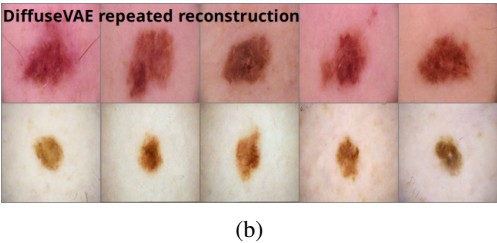

|                (a)                |                (b)                |

Figure 4: Repeated reconstructions from (a) cDDPM, and (b) DiffuseVAE. We notice that our cD-DPM has better shape consistency during reconstruction.

### 5.2.2 GUIDED RECONSTRUCTIONS WITH STYLE AND POSITION DECOMPOSITION

Our VAE can successfully decouple style and position parameters as demonstrated in figure 5. The first column gives the real images, and the second and the third columns are the corresponding multi-headed VAE and cDDPM reconstructions. Then, we switch the style parameters estimated by our VAE encoder between the two images, and the new VAE and cDDPM reconstructions are in the fourth and fifth columns. We see that our VAE decoder and cDDPM can successfully switch the styles of the images while still preserving lesion regions. As demonstrated, a user can combine position and style parameters from different images to generate images with a mixed appearance. In Appendix F we give further examples where we smoothly interpolate images by mixing style and position parameters.

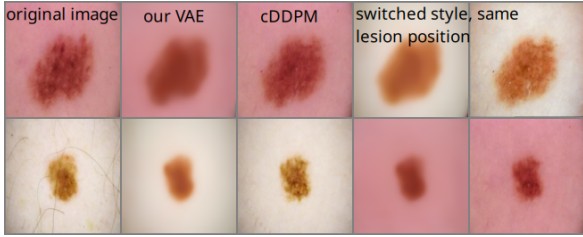

Figure 5: Reconstructions with switched style parameters.

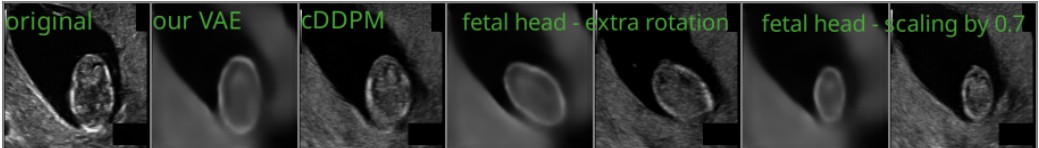

Figure 6: Reconstructions of a fetal image with guided position parameters.

When our VAE employs affine warp, users can easily alter the estimated affine matrix to guide image generation, as demonstrated in figure 6 by using a validation image from the FETAL dataset. The real fetal image is the first image, and the second and third images are the VAE and cDDPM reconstructions with estimated affine warp. After we obtain the affine matrix estimated by our VAE, we compose it with rotation by $\pi/4$ (fourth and fifth images) or scaling by $0.7$ (the last two images), where the center of the composed transformations are the centers of the fetal head. We notice that with the new position parameters, our new reconstructions only alter the fetal head, and the background images are mostly fixed. More examples of guided generations are in Appendix E.

## 6 CONCLUSION

We presented a framework for medical image segmentation and controlled image generation, which combines a multi-headed VAE and a conditional DDPM. Our multi-headed VAE enables a decomposition of the image style and position of object of interest, produces segmentation results comparable to state-of-the-art methods, and enables guided image reconstruction. When combined with our multi-headed VAE, our cDDPM can synthesize realistic images, where users can easily update or interpolate style and position estimates to provide extra guidance on the generation process.

Our approach certainly has limitations. For the segmentation task, we require that a common prior shape in 2D exists. However, many 2D segmentation tasks actually require a 3D shape prior, where the true masks are projections of a deformation of this 3D prior. It is more challenging to incorporate this setup in a VAE segmentation model. Another challenge is to explicitly model correlations between shape and some aspects of styles for medical images. For example, it is possible that certain types of skin lesions are more likely to have some specific texture and/or positions. Extracting this information from data and modeling it statistically requires further study.

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

## A  A STATISTICAL MODEL FOR IMAGE SEGMENTATION AND RECONSTRUCTION

In this section, we explain the statistical models underpinning the design of our multi-headed VAE in section 3. Given a medical image $\boldsymbol{x}$, we are interested in estimating a segmentation mask of the object, as well as estimating the style of the image (e.g. imaging modality, viewing angle, color tone, etc.). We call this the *image inference process*. On the other hand, given parameters describing positions of segmentation objects and image styles, we wish to generate a corresponding realistic image $\boldsymbol{x}$ with mask $\boldsymbol{y}$. We call this the *data generating process*. We will model both processes in a unified Bayesian approach.

### A.1  DATA GENERATING PROCESS

We assume the true location of the object can be described by a warping function $\mathrm{T}_{\boldsymbol{\theta}}$ parameterized by $\boldsymbol{\theta}$. We call $\boldsymbol{\theta}$ the *position parameter* and assume it follows the standard normal prior distribution. Given a sampled $\boldsymbol{\theta}$, a mask $\boldsymbol{y}$ is sampled by warping the reference $\boldsymbol{R}$ by $\mathrm{T}_{\boldsymbol{\theta}}$, i.e.

$$\boldsymbol{y}|\boldsymbol{\theta} \propto \exp\left(-\frac{\mathcal{U}(\boldsymbol{y}, \mathrm{T}_{\boldsymbol{\theta}}(\boldsymbol{R}))}{c}\right), \tag{16}$$

where $\mathcal{U}$ is a distance function, and $c$ is a strength parameter. Note that this functional form has also been used in (Van Leemput, 2008) for grid deformation analysis.

We assume the generating process of an image is governed by both the position parameter $\boldsymbol{\theta}$ and the style parameter $\boldsymbol{z}$ whose prior is also the standard normal distribution. Following the VAE approach in (Kingma & Welling, 2013), given both $\boldsymbol{\theta}$ and $\boldsymbol{z}$, we model the sampling of $\boldsymbol{x}$ by a *decoder neural network* $\mathrm{DEC}_{\boldsymbol{\phi}}$ with learnable parameters $\boldsymbol{\phi}$, i.e.

$$(\boldsymbol{\mu}, \boldsymbol{\sigma}) = \mathrm{DEC}_{\boldsymbol{\phi}}(\boldsymbol{\theta}, \boldsymbol{z}), \quad \boldsymbol{x}|(\boldsymbol{\theta}, \boldsymbol{z}) \overset{p_{\boldsymbol{\phi}}(\cdot|\boldsymbol{\theta}, \boldsymbol{z})}{\sim} \mathcal{N}(\boldsymbol{\mu}, \mathrm{diag}(\boldsymbol{\sigma}^2)). \tag{17}$$

### A.2  POSTERIOR ESTIMATION

For an observed image $\boldsymbol{x}$ we follow the standard VAE approach as in subsection 2.1 to approximate the posterior probability $p_{\boldsymbol{\phi}}(\boldsymbol{\theta}, \boldsymbol{z}|\boldsymbol{x})$ by an encoder network $\mathrm{ENN}_{\boldsymbol{\psi}}(\boldsymbol{x})$, where now the pair $(\boldsymbol{\theta}, \boldsymbol{z})$ plays the role of the latent variable $\boldsymbol{z}$ in section 2.1. Now, the new ELBO function is given by

$$\mathcal{L}(\boldsymbol{\phi}, \boldsymbol{\psi}|\boldsymbol{x}) = \mathbb{E}_{q_{\boldsymbol{\psi}}(\boldsymbol{\theta}, \boldsymbol{z}|\boldsymbol{x})}\left[\log p_{\boldsymbol{\phi}}(\boldsymbol{x}|\boldsymbol{\theta}, \boldsymbol{z})\right] - D_{\mathrm{KL}}(q_{\boldsymbol{\psi}}(\boldsymbol{\theta}, \boldsymbol{z}|\boldsymbol{x})\|p(\boldsymbol{\theta}, \boldsymbol{z})), \tag{18}$$

where $p(\boldsymbol{\theta}, \boldsymbol{z})$ is the joint prior distribution of $(\boldsymbol{\theta}, \boldsymbol{z})$ (we assume $\boldsymbol{\theta}$ and $\boldsymbol{z}$ are independent), and we denote the distribution induced by this approximate by $q_{\boldsymbol{\psi}}(\boldsymbol{\theta}, \boldsymbol{z}|\boldsymbol{x})$.

For segmentation training dataset, we always observe paired data $(\boldsymbol{x}, \boldsymbol{y})$. Observing the image $\boldsymbol{x}$ tells us information about both the style $\boldsymbol{z}$ and the position parameter $\boldsymbol{\theta}$, while observing the mask tells us information about $\boldsymbol{\theta}$ through the posterior distribution $p(\boldsymbol{\theta}|\boldsymbol{y})$. Since the pair $(\boldsymbol{x}, \boldsymbol{y})$ share the same position parameter, the two posteriors $p_{\boldsymbol{\phi}}(\boldsymbol{\theta}|\boldsymbol{x})$ and $p_{\boldsymbol{\phi}}(\boldsymbol{\theta}|\boldsymbol{y})$ should be close. Therefore another way to ensure $q_{\boldsymbol{\psi}}(\boldsymbol{\theta}|\boldsymbol{x})$ approximates $p_{\boldsymbol{\phi}}(\boldsymbol{\theta}|\boldsymbol{x})$ is to minimize

$$
\begin{aligned}
&D_{\mathrm{KL}}(q_{\boldsymbol{\psi}}(\boldsymbol{\theta}|\boldsymbol{x})\|p(\boldsymbol{\theta}|\boldsymbol{y})) \\
&= \int q_{\boldsymbol{\psi}}(\boldsymbol{\theta}|\boldsymbol{x}) \log \frac{q_{\boldsymbol{\psi}}(\boldsymbol{\theta}|\boldsymbol{x})}{p(\boldsymbol{\theta}|\boldsymbol{y})} \mathrm{d}\boldsymbol{\theta} \\
&= \int q_{\boldsymbol{\psi}}(\boldsymbol{\theta}|\boldsymbol{x}) \log \frac{q_{\boldsymbol{\psi}}(\boldsymbol{\theta}|\boldsymbol{x})p(\boldsymbol{y})}{p(\boldsymbol{\theta}, \boldsymbol{y})} \mathrm{d}\boldsymbol{\theta} \\
&= \log p(\boldsymbol{y}) + \int q_{\boldsymbol{\psi}}(\boldsymbol{\theta}|\boldsymbol{x}) \log \frac{q_{\boldsymbol{\psi}}(\boldsymbol{\theta}|\boldsymbol{x})}{p(\boldsymbol{y}|\boldsymbol{\theta})p(\boldsymbol{\theta})} \mathrm{d}\boldsymbol{\theta} \\
&= \log p(\boldsymbol{y}) + \int q_{\boldsymbol{\psi}}(\boldsymbol{\theta}|\boldsymbol{x}) \log \frac{q_{\boldsymbol{\psi}}(\boldsymbol{\theta}|\boldsymbol{x})}{p(\boldsymbol{\theta})} \mathrm{d}\boldsymbol{\theta} - \int q_{\boldsymbol{\psi}}(\boldsymbol{\theta}|\boldsymbol{x}) \log p(\boldsymbol{y}|\boldsymbol{\theta}) \mathrm{d}\boldsymbol{\theta} \\
&= \log p(\boldsymbol{y}) + \underbrace{D_{\mathrm{KL}}(q_{\boldsymbol{\psi}}(\boldsymbol{\theta}|\boldsymbol{x})\|p(\boldsymbol{\theta})) - \mathbb{E}_{q_{\boldsymbol{\psi}}(\boldsymbol{\theta}|\boldsymbol{x})}[\log p(\boldsymbol{y}|\boldsymbol{\theta})]}_{:=-\widehat{\mathcal{L}}(\boldsymbol{\psi}|\boldsymbol{x}, \boldsymbol{y})},
\end{aligned}
$$

which is equivalent to maximizing another ELBO function

$$\widehat{\mathcal{L}}(\boldsymbol{\psi}|\boldsymbol{x},\boldsymbol{y}) = \mathbb{E}_{q_{\boldsymbol{\psi}}(\boldsymbol{\theta}|\boldsymbol{x})}[\log p(\boldsymbol{y}|\boldsymbol{\theta})] - D_{\mathrm{KL}}(q_{\boldsymbol{\psi}}(\boldsymbol{\theta}|\boldsymbol{x})\|p(\boldsymbol{\theta})). \tag{19}$$

Note that if we use the Dice score as the distance function $\mathcal{U}$ in equation 16, then maximizing this new ELBO function corresponds to minimizing the Dice score for mask prediction.

Combining equation 18 and equation 19, to train our decoder and encoder jointly we maximize

$$\mathbb{E}_{(\boldsymbol{x},\boldsymbol{y})\sim p_{\mathrm{data}}}[\mathcal{L}(\boldsymbol{\phi},\boldsymbol{\psi}|\boldsymbol{x}) + \widehat{\mathcal{L}}(\boldsymbol{\psi}|\boldsymbol{x},\boldsymbol{y})]. \tag{20}$$

This explains our choice of the loss function in section 3.4, with the exception that now the discrepancy between the reconstructed image and original image is measured by squared $L_2$ norm. We note that as observed in (Chen et al., 2019), using the log-cosh function instead of the squared $L_2$ norm can give better reconstruction, and in our actual implementation, we use the log-cosh function as the reconstruction loss.

## B    SOLUTION OF THIN PLATE SPLINE COEFFICIENTS

In this section, we explain how the coefficients of the TPS transformation in equation 12 are calculated. We use the $x$-coordinate coefficients in equation 12a as an example, and the calculation of the $y$ coordinate coefficients is done in a similar manner.

The thin plate function 12a has $K + 3$ coefficients to be computed. Though the function is highly non-linear with the kernel function $U$, the function is linear with respect to the coefficients. Hence, the coefficients can be computed explicitly. We adopt notations defined in section 3.2.2.

Let $\mathbf{v} = (x'_1, \cdots, x'_K|0,0,0)^T$, where $x'_i$ is the $x$-coordinate of the $i$-th source control point. Also, define matrices

$$\mathcal{K} = \begin{bmatrix} 0 & U_{12} & \cdots & U_{1K} \\ U_{21} & 0 & \cdots & U_{2K} \\ \cdots & \cdots & \cdots & \cdots \\ U_{K1} & U_{K2} & \cdots & 0 \end{bmatrix}, K \times K; \quad \mathcal{P} = \begin{bmatrix} 1 & x_1 & y_1 \\ 1 & x_2 & y_2 \\ \cdots & \cdots & \cdots \\ 1 & x_K & y_K \end{bmatrix}, K \times 3; \tag{21}$$

and

$$\mathcal{M} = \begin{bmatrix} \mathcal{K} & \mathcal{P} \\ \mathcal{P}^T & O \end{bmatrix}, (K+3) \times (K+3) \tag{22}$$

where $U_{i,j} = U(|\mathbf{p}_i - \mathbf{p}_j|)$, $x_i$ and $y_i$ are the $x$- and $y$-coordinates of targart control point $\mathbf{p}_i$, and $O$ is a zero matrix of size $3 \times 3$. Then the coefficients $a_k$'s are given by

$$\mathbf{a} = (a_1, \cdots, a_{K+1})^T = \mathcal{M}^{-1}\mathbf{v}. \tag{23}$$

The additional last three rows of $\mathcal{M}$ guarantee that the coefficients $a_k$ sum to zero and that their cross-products with the points $\mathbf{p}_i$ are likewise zero.

In our multi-headed VAE implementation, the target control points $\mathbf{p}_i$'s are fixed. Therefore, the inverse of $\mathcal{M}$ only needs to be calculated once and can be cached, and it only requires one matrix-vector multiplication to calculate new coefficients $\mathbf{a}$ for updated source control points.

## C    CHOICE OF THIN PLATE SPLINE CONTROL POINTS

The number and placement of control points in TPS deformation are not fixed. In an extreme scenario, one can designate every pixel location as a control point. In practice, however, having a few control points at key locations is sufficient to achieve a realistic deformation effect. Using fewer control points is computationally advantageous since it reduces the size of the inverse matrix in Equation 23 and the number of source control points the model has to estimate.

The figure 7 displays the deformation results using two control point configurations, namely uniform and circular, for deforming two merged circles into a single circle. The uniform and circular configurations use 25 and 20 control points, respectively. The results demonstrate that both configurations can successfully approximate the target shape. However, we observe that the learned grid in the uniform configuration exhibits a significantly more non-linear behavior than the circular configuration. Furthermore, most of the control points located in the background appear to be redundant.

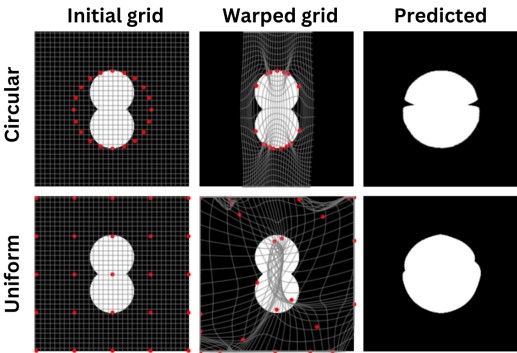

Figure 7: TPS deformation results from two control point configurations, uniform and circular, for deforming two merged circles into a single circle. The gray lines represent the coordinate grid, and the red dots represent the target control points.

Based on these observations, we hypothesize that employing a circular configuration in real applications may lead to more generalized segmentation results by reducing excessively non-linear deformations and also ensuring that more control points contribute to warping the regions containing objects. The performance of the two control point configurations in segmenting skin lesions from the ISIC_2018 dataset is illustrated in Table 2. We notice a significantly improved performance with the circular configuration in both Hausdorff and Dice measures.

| Control point | HD($\downarrow$)$\pm$sd | Dice($\uparrow$)$\pm$sd |
|---|---|---|
| uniform | $16.43 \pm 0.54$ | $0.89 \pm 0.00$ |
| circular | $15.47 \pm 0.65$ | $0.90 \pm 0.00$ |

Table 2: Performance comparison between uniform and circular control point configurations on ISIC_2018.

## D  DATASET

We tested segmentation on three publicly available datasets:

1. ISIC_2018 (Codella et al., 2019): 2,594 dermoscopic images and segmentation of lesion regions, evaluated using 5-fold cross-validation,

2. RIM-ONE-DL (Batista et al., 2020): 485 (311 training, 174 testing) retinographies from normal and glaucoma patients, with segmentation masks of optic disc and optic cup. We evaluated both tasks on the train-test split,

3. FETAL (van den Heuvel et al., 2018): 999 ultrasound images and segmentation masks of the standard plane of fetal heads. We evaluated using 5-fold cross-validation.

We tested our cDDPM on ISIC and FETAL datasets. For ISIC, we used the VAE trained on the first fold of ISIC_2018 and trained cDDPM on ISIC_2020 (Rotemberg et al., 2021). ISIC_2020 contains 33,126 images without segmentation masks, and we used a $85\%$-$15\%$ train-validation split. For FETAL, we used the VAE trained on the first fold and trained cDDPM on the training set of the first fold. When training cDDPM, we keep our VAE fixed.

## E  RECONSTRUCTIONS WITH PERTURBED AFFINE WARPS

In figure 8, we give more examples of guided generations with user-provided position alterations. First row: real images from FETAL validation set. Second and third rows: our VAE and cDDPM reconstructions using estimated affine warps. Fourth and fifth rows: extra rotation of fetal heads by $\pi/4$. Last two rows: scaling of fetal heads by $0.7$. Both extra rotation and scaling are done with

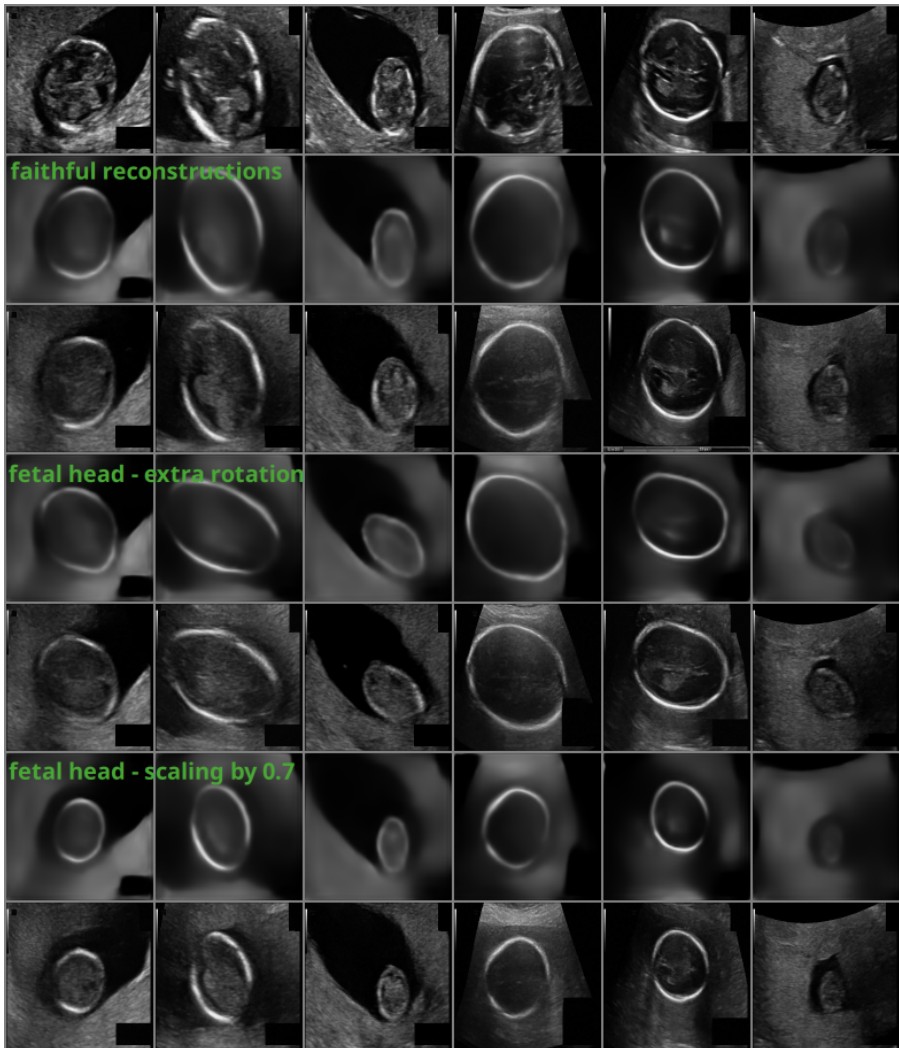

Figure 8: Reconstruction of fetal validation images with guided position parameters.

respect to the center of fetal heads. We notice that by only altering the position parameters, our VAE and cDDPM can preserve the style of images (textures, image backgrounds) while changing the fetal heads accordingly.

## F  CONDITIONAL IMAGE INTERPOLATION

In figure 9, we demonstrate smooth interpolation of two images (top left and bottom right images) where we preserve either the image style or the lesion shape/position. For each row in the sub-figures, we gradually interpolate the styles (mostly the color tone and skin texture) of the images while keeping the lesion region. This is done by generating images with interpolated style parameters with a fixed position parameter. For each column, we change the lesion region while keeping the style in a similar manner. When generating images in each sub-figure, we use a fixed sequence of isotropic Gaussian noise when solving the backward stochastic differential equation 15.

We note that our approach can mostly preserve either image style or lesion region during interpolations. However, due to the randomness in the backward diffusion process, some tiny features (e.g. hair) cannot always be smoothly interpolated.

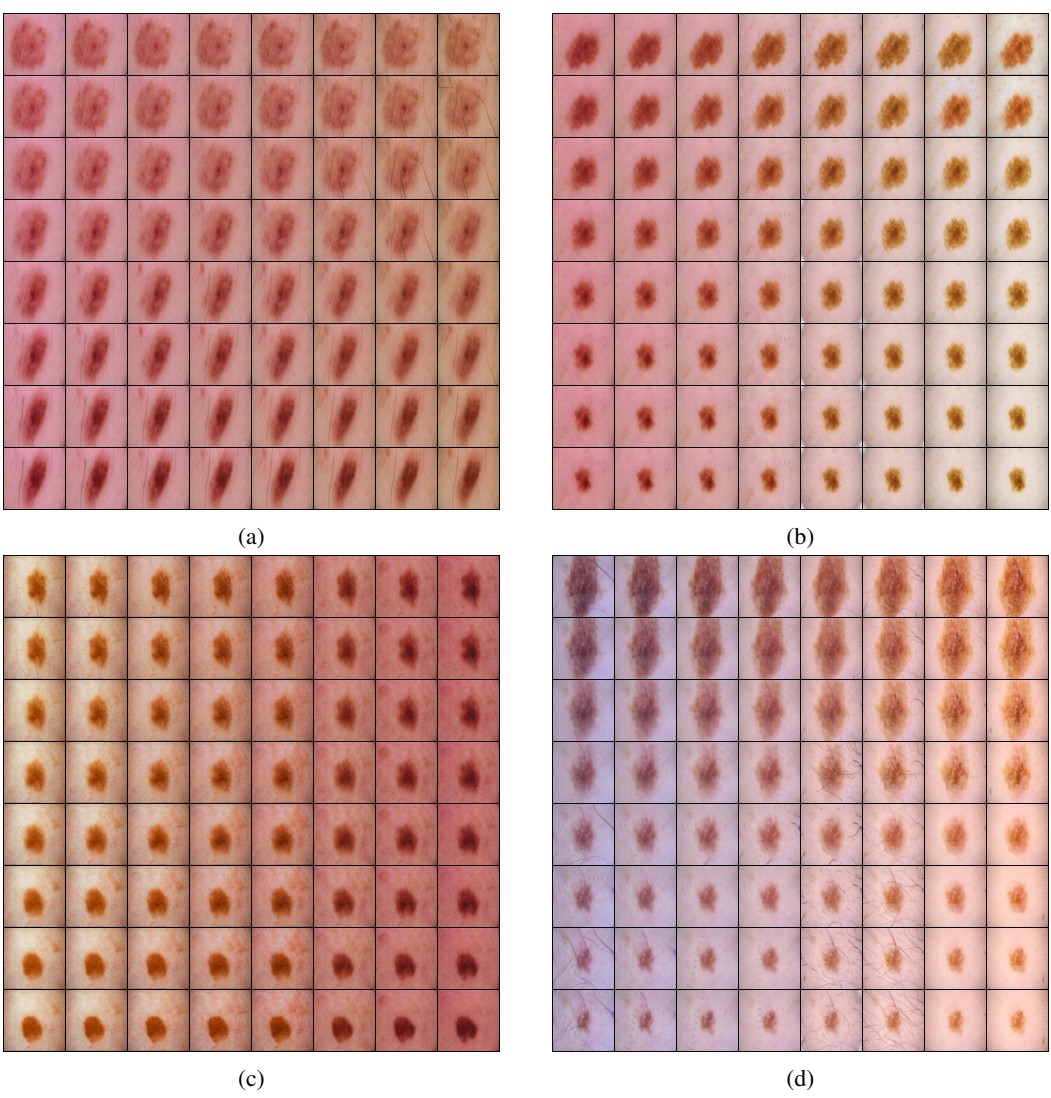

(a) (b)

(c) (d)

Figure 9: Smooth interpolation of images with preserved styles or lesion regions. Rows: interpolated styles with fixed position. Columns: interpolated posiitons with fixed style.

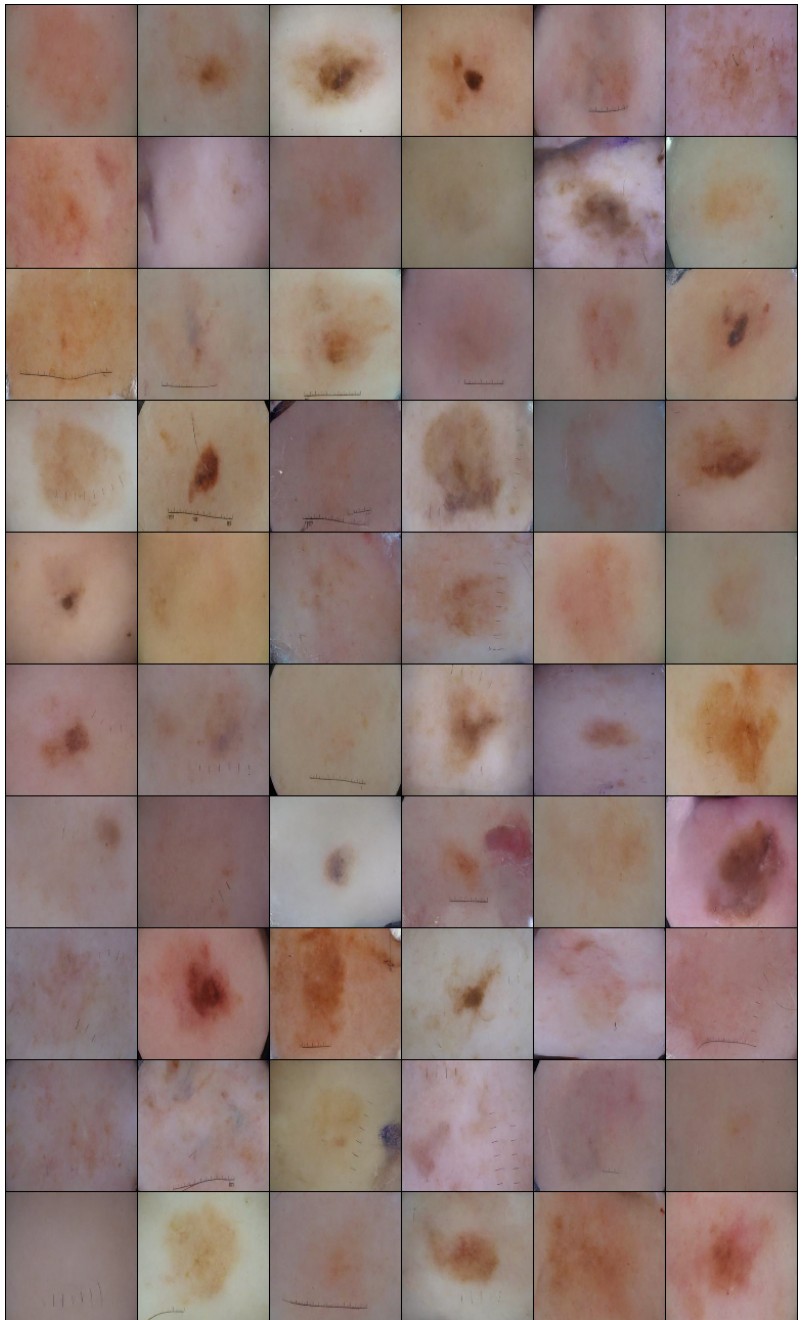

Figure 10: Pure sampling by cDDPM.

## G    PURE SAMPLING BY CDDPM

Instead of controlled image generation, our cDDPM can also perform pure sampling with samples shown in figure 10. Based on 5000 sampled images, our method achieves an FID score of 79.39 for skin lesion image generation. For comparison, the DiffuseVAE (Pandey et al., 2022) trained in similar hyper-parameters can achieve an FID score of 83.81. This shows that our proposed method not only has more controllable image generation but can also generate realistic and diverse images.

