# OpenReview forum: "Enhancing Medical Image Generation with Anatomical Precision: A Multi-Headed VAE-Based Diffusion Model"
_ICLR.cc/2024/Conference — ICLR 2024 Conference Withdrawn Submission_

### Official Review · Reviewer_PEau · 2023-10-28

**Soundness:** 2 fair
**Presentation:** 2 fair
**Contribution:** 2 fair
**Rating:** 3
**Confidence:** 4

**Summary:**

This paper proposed a unified framework for image segmentation and generation, where the segmentation is done by a VAE, and the generation is done by a VAE and Diffusion refinement.

**Strengths:**

1. Unify the segmentation and image reconstruction into a single VAE model with proposed location and style decomposition.
2. Propose a conditional denoising diffusion probabilistic model (cDDPM) to refine the image from the VAE.

**Weaknesses:**

1. The author proposed two types of spatial transformation, one is affine transform and the other is thin plate spline transform. Both types of transform only work well on simple segmentation like skin disease. I highly recommend the author test on some more complex anatomical organs, like the pancreas or cardiac ultrasound.
2. In the image reconstruction of the VAE, the author only warped the first half of the channel using the deformation from the location branch. If the shape and style are fully decomposed, they should warp the entire features without the skip connection. Otherwise, the following decoder block might ignore the deformed feature and only utilize the skipped feature. Therefore, I want to author to add an experiment showing the performance of reconstruction without the skip connection.
3. For the diffusion refinement, I'm not sure why the author wants to utilize such a complex framework. According to my experience, if you simply utilize the conditional DDPM by concatenating the noisy input with the blurred image from VAE, you're able to train a CDDPM following the traditional DDPM training framework. I'd like to see this extra experiment.
4. I don't think this is a general approach for the medical images. If so, the author should utilize more popular datasets in the image segmentation from multi-domains.

**Questions:**

Please add experiments and answer my question in the weakness part.
Furthermore, I have some other questions:
1. If this work is for the generation, I'd like to see some more quantitative results, like LPIPS and FID.
2. What is the use case and benefit of this image generation framework? I don't get a clear motivation from the author.

**Details Of Ethics Concerns:**

No ethics concerns.

---

### Official Review · Reviewer_nez1 · 2023-11-01

**Soundness:** 3 good
**Presentation:** 4 excellent
**Contribution:** 2 fair
**Rating:** 5
**Confidence:** 4

**Summary:**

In this work, the authors aim to disentangle position and style in variational autoencoders (VAE). This is so that the user can easily choose to change either the position or the style (eg: color) while generating medical images. To overcome the issue of blurry images produced by VAE, the authors also use diffusion models to enhance the quality of the image. The authors’ method generates both an image and its corresponding ground truth segmentation mask. The segmentation task helps in the disentanglement process. The authors validate the segmentation performance on three datasets. They provide qualitative results showing that they are able to achieve disentanglement between position and style.

**Strengths:**

1) The authors have a clever design of injecting the position posterior into the intermediate layer in order to alleviate the style parameter from learning the position features.
2) Through visual qualitative results, the authors demonstrate that they are able to disentangle the position and style parameters, and generate appropriate results by keeping one fixed and changing the other.
3) The authors provide clear proof of their selection of loss function.

**Weaknesses:**

1) Since the authors’ main contribution is disentanglement, the authors should consider providing a comparison to or atleast a discussion of controllable generative models like ControlVAE [1], DynamicVAE [2], ControlNet [3]. The authors should provide comparison if possible, or, provide a discussion on how the mentioned methods are different / not applicable.
2) One of the main limitations of this method is the need for a good reference mask. Could the authors provide a discussion on how their method would behave in the case of instance segmentation (for datasets like GLaS [4]) and curvilinear segmentation (for datasets like DRIVE [5])?
3) It is unclear if the comparison of segmentation performance is fair or not. For non-generative models like UNet++, are the numbers on the test set? While for generative models like the proposed cDDPM, are the numbers on the generated images?


**References**

[1] Shao, Huajie, et al. "Controlvae: Controllable variational autoencoder." International Conference on Machine Learning. PMLR, 2020.

[2] Shao, Huajie, et al. "Dynamicvae: Decoupling reconstruction error and disentangled representation learning." arXiv preprint arXiv:2009.06795 (2020).

[3] Zhang, Lvmin, Anyi Rao, and Maneesh Agrawala. "Adding conditional control to text-to-image diffusion models." Proceedings of the IEEE/CVF International Conference on Computer Vision. 2023.

[4] Sirinukunwattana, Korsuk, et al. "Gland segmentation in colon histology images: The glas challenge contest." Medical image analysis 35 (2017): 489-502.

[5] Staal, Joes, et al. "Ridge-based vessel segmentation in color images of the retina." IEEE transactions on medical imaging 23.4 (2004): 501-509.

**Questions:**

1) Please also see the weakness above.
2) The authors state, “When our VAE employs affine warp, users can easily alter the estimated affine matrix”. Since the authors use TPS as well, how should the users alter the position in TPS?
3) In Section G, the authors mention that their method can also do pure sampling. How is this achieved, that is, how does the procedure change from controlled to pure sampling?
4) In Figure 10, the pure sampling does not capture the lesion area well (which the authors state as a limitation of DiffuseVAE earlier). Please discuss.
5) Please mention if the numbers in bold are just numerically better, or, if t-test [6] has been conducted to check if the performance improvement is statistically significant or not.

**References**

[6] Student, 1908. The probable error of a mean. Biometrika, pp.1–25.

---

### Official Review · Reviewer_etX6 · 2023-11-02

**Soundness:** 2 fair
**Presentation:** 2 fair
**Contribution:** 2 fair
**Rating:** 5
**Confidence:** 5

**Summary:**

This article proposes a new diffusion model that guides the reverse diffusion process by using a specially designed multi head VAE, which generates images that display true style and anatomical accuracy. Solved the problem of difficulty in accurately capturing global anatomical priors in diffusion models and the lack of control over the ability to convert unrealistic anatomical images into real images due to recalibration.The proposed method has shown good performance in a series of medical image tasks, such as skin damage and fetal head. In addition, the model provides state-of-the-art segmentation performance.

**Strengths:**

1.The Conditional DDPM (cDDPM) proposed in this article combines multi head VAE and continuous conditional diffusion models. It can effectively refine the blurred VAE reconstruction and control the generation of real images, allowing for more faithful reconstruction of image styles and segmented regions.
2.The model proposed in this article decouples meaningful potential representations of style and position, and manipulating potential parameters of style or position only affects the corresponding parts of the generated image.
3.This article proposes a new VAE model for image segmentation and generation, which achieves optimal performance compared to other advanced methods.

**Weaknesses:**

1.The methods proposed in this article can mostly preserve image style or lesion areas during the interpolation process. However, the generation effect of certain small features (such as hair) is poor.
2. limited novelty. it seems a combination of VAE and diffusion.

**Questions:**

1.Why didn't ablation experiments be conducted, and how did the effectiveness of each module be verified?
2.Why is there a problem of small feature loss? How to solve it?
3.Why not compare the segmentation results with the latest article? Why is there no comparative experiment on the synthesis results?